# Cinnamaldehyde Inhibits Postharvest Gray Mold on Pepper Fruits via Inhibiting Fungal Growth and Triggering Fruit Defense

**DOI:** 10.3390/foods12183458

**Published:** 2023-09-16

**Authors:** Lifei Yang, Xiaoli Liu, Haiyan Lu, Cunzheng Zhang, Jian Chen, Zhiqi Shi

**Affiliations:** 1Hexian New Countryside Development Research Institute, College of Horticulture, Nanjing Agricultural University, Nanjing 210095, China; lfy@njau.edu.cn (L.Y.); 2017104077@njau.edu.cn (X.L.); 2Laboratory for Food Quality and Safety-State Key Laboratory Cultivation Base of Ministry of Science and Technology, Institute of Food Safety and Nutrition, Jiangsu Academy of Agricultural Sciences, Nanjing 210014, China; haiyanlu@jaas.ac.cn (H.L.); zcz@jaas.ac.cn (C.Z.); chenjian@jaas.ac.cn (J.C.)

**Keywords:** antifungal, *Botrytis cinerea*, cinnamaldehyde, gray mold, induced defense, pepper fruit

## Abstract

Gray mold infected with *Botrytis cinerea* frequently appears on fruits and vegetables throughout the supply chain after harvest, leading to economic losses. Biological control of postharvest disease with phytochemicals is a promising approach. CA (cinnamaldehyde) is a natural phytochemical with medicinal and antimicrobial activity. This study evaluated the effect of CA in controlling *B. cinerea* on fresh pepper fruit. CA inhibited *B. cinerea* growth in vitro significantly in a dose- (0.1–0.8 mM) and time-dependent (6–48 h) manner, with an EC_50_ (median effective concentration) of 0.5 mM. CA induced the collapse and breakdown of the mycelia. CA induced lipid peroxidation resulting from ROS (reactive oxygen species) accumulation in mycelia, further leading to cell leakage, evidenced by increased conductivity in mycelia. CA induced mycelial glycerol accumulation, resulting in osmotic stress possibly. CA inhibited sporulation and spore germination resulting from ROS accumulation and cell death observed in spores. Spraying CA at 0.5 mM induced a defense response in fresh pepper fruits, such as the accumulation of defense metabolites (flavonoid and total phenols) and an increase in the activity of defense enzymes (PAL, phenylalanine ammonia lyase; PPO, polyphenol oxidase; POD, peroxidase). As CA is a type of environmentally friendly compound, this study provides significant data on the activity of CA in the biocontrol of postharvest gray mold in peppers.

## 1. Introduction

Postharvest disease development is a major cause of food waste and economic losses worldwide [1]. *Botrytis cinerea* is the main phytopathogenic fungi causing postharvest gray mold on most fresh fruits and vegetables [2]. Infection with *B. cinerea* results in decay and rot on fruits, leading to a decline in quality and economic losses. *B. cinerea* can survive easily in variable environmental conditions, producing spores than can be easily dispersed to invade neighboring fruits and vegetables [3]. Pepper is a widely cultivated vegetable that is economically important for fresh consumption and processing worldwide. *B. cinerea*, causing gray mold, significantly decreases the fruit quality during the transport and storage of pepper [4].

Controlling postharvest gray mold has been considered a great challenge in avoiding the considerable postharvest losses of fruits and vegetables. Several approaches have been developed to control postharvest gray mold. The first one is a controlled atmosphere and temperature. *B. cinerea* can survive and cause disease at a wide range of temperatures (0–26 °C), but this pathogen can grow and infect rapidly under favorable environmental conditions (18–24 °C; RH > 93%) [5,6,7]. A low temperature and dry environment can slow down the development and spread of *B. cinerea* during the storage of fruits and vegetables. The fungi can grow rapidly in the environment with an enhanced temperature and humidity. Therefore, the temperature and humidity should be precisely controlled during cold-chain transportation, which may increase the cost of long-distance transportation [8]. The second approach is heat treatment. Rinsing bell peppers with hot water at 55 °C for 12 s can disinfect peppers and prolong storage by killing pathogens and altering the host fruit physiology. However, this approach relies on specific equipment for hot water rinsing and brushing [9]. The third approach is using physical means, such as ozone, UV irradiation, or altered atmospheric conditions (hypobaric or hyperbaric treatments). These approaches can control postharvest gray mold effectively without direct contact with the fruit, but the effect always lasts for a short time [8]. The fourth approach is using fungicides. Chemical fungicides can kill the pathogens directly, but toxic fungicide residues may pose a threat to human health. Moreover, using a single fungicide for a long time may drive the occurrence of drug resistance in *B. cinerea*. Some countries even restrict the usage of conventional synthetic fungicides in postharvest disease management [8,10]. The fifth approach is the biological control of postharvest disease, which has been drawing great attention recently [11]. Among the biological methods, the use of environmentally friendly natural compounds (e.g., essential oils, plant extracts, and microbial products) is a promising approach [4,12]. These natural compounds have fungicidal activity against *B. cinerea*, but they are safe to human health and the environment. Some natural compounds even have the ability to induce resistance in fruits and vegetables, limiting the development of postharvest disease [13].

CA (cinnamaldehyde) is a phytochemical obtained from *Cinnamomum* species [14]. CA has antimicrobial and immune-regulatory activity [15,16]. CA has been applied as a potential natural food preservative due to its antifungal and antibacterial activity [17,18]. CA (3-phenylpropenal, C_9_H_8_O) is a type of aldehyde with one phenyl ring (Appendix A). The fungicidal activity of CA has been associated with the conjugated double bond and the length of the CH chain outside the ring [19]. The fungicidal activity of CA against plant pathogens has been documented [20,21], and the crude extracts of cinnamon can inhibit gray mold on strawberry [22]. However, we still have little knowledge about the role of CA against gray mold caused by *B. cinerea* and its mechanisms.

In this work, we studied the effect of CA on the control of gray mold on fresh pepper. First, the antifungal activity of CA against *B. cinerea* was investigated. Then, CA-induced defensive responses in pepper fruits were detected. Finally, the possible mechanisms and their significance were discussed.

## 2. Materials and Methods

### 2.1. Strain, Media, and Treatment

The monoculture of *B. cinerea* was obtained from the College of Plant Protection in Nanjing Agricultural University, China. *B. cinerea* were placed in PDA (potato dextrose agar) medium in a petri dish at 25 °C and in PDB (potato dextrose broth) medium in a shaker (150 rpm) at 25 °C. Both PDA and PDB were obtained from Nanjing Rongshengda Instrument Co., Ltd., Nanjing, China.

CA of analytical grade (>98%) was obtained from Sinopharm Chemical Reagent Co., Ltd., Shanghai, China. CA at different concentrations (0, 0.1, 0.2, 0.4, 0.6, and 0.8 mM) was added to PDA shortly before the solidification of PDA. Then, the PDA containing CA was transferred to petri dishes for the culture of *B. cinerea* for 48 h. Then, we measured the colony diameter using a ruler crossing the center of the colony circle to evaluate the radical growth of *B. cinerea* [23]. 

The EC_50_ (median effective concentration) of CA against *B. cinerea* was 0.5 mM based on the calculation of the colony diameter determined as above (linear regression) [24]. The colony diameter was also determined at 6, 12, 24, 36, and 48 h, respectively, under the treatment of CA at 0.5 mM. The average growth speed of *B. cinerea* was calculated using linear regression based on the changes in colony diameter in 48 h. Then, CA at 0.2, 0.5, and 0.8 mM (representing low, median, and high CA concentrations, respectively) was added to PDB for the culture of *B. cinerea* to collect mycelia for physiological measurements.

### 2.2. Determination of Mycelial Biomass

*B. cinerea* was cultured in PDB containing CA for 48 h. Then, the mycelia were collected through filter paper. The mycelia were surface-dried with paper tissue gently. The fresh weights of mycelia was determined using a laboratory balance. Then, the fresh mycelia were dried in an oven (60 °C) for 48 h, followed by measuring the mycelial dry weight [25]. Three replicates were conducted for each treatment.

### 2.3. Spore Production and Germination

The spores were obtained from a PDA-cultured strain according to a previously published method with slight modification [26]. *B. cinerea* was cultured in PDA in a petri dish (2 cm in diameter) for 10 days to produce sufficient spores. Then, sterilized water (2 mL) was added to the petri dish to wash the spores out. The spores were collected through four layers of filter paper followed by centrifugation at 10,000× *g* for 5 min.

CA at different concentrations (0.2, 0.5, and 0.8 mM) was added to PDA for the culture of *B. cinerea* for 6 days to produce spores. All the spores were collected to count the number of spores produced. 

To test the inhibitory effect of CA on spore germination, the spore suspension (1 × 10^5^ spores/mL) was cultured on agar medium containing CA at 0.2, 0.5, and 0.8 mM, respectively, for 12 h. Then, the number of germinated spores was counted for the calculation of the germination rate.

### 2.4. SEM (Scanning Electron Microscopy)

The mycelia of *B. cinerea* were collected by filtration of the culture solution with double gauze, followed by washing with phosphate buffer (0.2 M, pH 7.0). Then, we fixed the mycelia with graded ethanol and acetone according to our previously published method [23]. Then, the mycelial samples were observed with an SEM (EVO-LS10, ZEISS, Jena, Germany). The structure of mycelia was evaluated based on morphological analysis [27].

### 2.5. Meaaurement of Mycelial Conductivity

The mycelia (0.5 g) of *B. cinerea* after treatment (48 h) were collected and suspended in 20 mL distilled water. Then, a conductivity meter (CON510 Eutech/Oakton, Singapore) was applied to measure the electrical conductivity at 0, 5, 10, 20, 40, 60, 80, 100, 120, 140, 160, and 180 min, respectively. The final conductivity was measured after incubating the mycelia in boiled water for 5 min. The relative conductivity was calculated as follows:$$\mathrm{Relative}\;\mathrm{conductivity}\;(\%)=\frac{\mathrm{Conductivity}}{\mathrm{Final}\;\mathrm{conductivity}}\times 100$$

### 2.6. Meaaurement of Mycelial TBARS (Thiobarbituric Acid Reactive Substances) Content

The mycelia after treatment were collected by filtration of the culture solution with double gauze, followed by measuring the TBARS content using a detection kit (A003; Nanjing Jiancheng Bioengineering Institute, Nanjing, China) [28].

### 2.7. Measurement of Mycelial Glycerol Content

Approximately 0.5 g of mycelia after treatment were harvested. The mycelia were ground and suspended with 20 mL of distilled water. The mixture was water-heated at 80 °C for 15 min, followed by centrifuging (8500 rpm, 10 min). CuSO_4_ (0.05 g/L) was mixed with the supernatant, with shaking at 100 rpm for 12 min. This allowed the reaction between glycerol and CuSO_4_ to produce a dark blue copper–glycerol complex with specific absorbance at 630 nm. Then, the absorbance at 630 nm was recorded for the filtered mixture. The glycerol in mycelial samples was calculated based on a standard curve of glycerol (0–0.01 g/mL) [29].

### 2.8. Histochemical Analysis of ROS (Reactive Oxygen Species) and Cell Death

The mycelia after treatment with CA for 48 h were washed with distilled water three times. The mycelia were stained with 1 µM DCFH-DA (2′,7′-dichlorofluorescein diacetate) at 25 °C for 20 min. The endogenous ROS were labeled with DCF fluorescence (green) and analyzed by fluorescent microscopy (ECLIPSE Series, TE2000-S, Nikon, Tokyo, Japan) [30].

For the detection of cell death, collected mycelia were stained with 5 µM PI (propidium iodide) at 25 °C for 30 min. The dead cells were labeled with PI fluorescence (red) and photographed under fluorescent microscopy [31].

The spores of *B. cinerea* were also incubated with DCFH-DA or PI, followed by fluorescence assays using flow cytometry (Accuri C6 Plus, BD Biosciences, San Jose, CA, USA) [32].

### 2.9. CA-Suppressed Development of B. cinerea on Pepper Fruits

Pepper seeds (Sujiao 5) were obtained from the Institute of Vegetable Crops at Jiangsu Academy of Agricultural Sciences, Nanjing, China. The peppers were grown in a greenhouse without using any pesticides. The harvested pepper fruits were collected to test the effect of CA on the control of *B. cinerea* development. Fresh pepper fruits were surface-sterilized with 0.5% NaClO for 5 min and washed with sterilized water three times. An agar plug (5 mm in diameter) taken from PDA-cultured *B. cinerea* was inoculated on the surfaces of pepper fruits. Each fruit was inoculated with five mycelial plugs with an even distribution on the fruit surface (Appendix A). The fruits were kept in a light chamber (photoperiod of 12 h and active radiation of 200 μmol/(m^2^ s)) at 25 °C, allowing the growth and infection of *B. cinerea*. Pepper fruits were sprayed with CA at 0.5 mM for 2 h before inoculating *B. cinerea*. The peppers in the control group were treated with sterilized water before inoculation. Five fruits were inoculated for each treatment. The sizes of disease lesions were measured after 7 days. For the size of each lesion (almost a circle), the diameter was evaluated by measuring it three times from different directions crossing the center of the lesion.

### 2.10. Assay of Defensive Enzyme Activity of Pepper Fruits

Approximately 2 g pepper fruit samples around the inoculated sites were taken for the measurement of enzyme activity. The fruit samples were homogenized with pre-chilled PBS (phosphate buffer solution, 50 mM, pH 7.0) containing 5 mM β-mercaptoethanol and 2 mM EDTA. After centrifugation at 12,000 rpm at 4 °C for 20 min, the supernatant was collected as extracts for the determination of the enzyme activity.

The POD (peroxidase) activity was measured based on guaiacol oxidation with hydrogen peroxide [33]. The reaction mixture (2 mL) consisted of PBS, 0.5% hydrogen peroxide, 0.5% guaiacol, and 20 µL of extract. Then, we recorded the absorbance at 420 nm for the calculation of POD activity using the extinction coefficient at 26.6 /mM/cm.

The PPO (polyphenol oxidase) activity was measured based on pyrocatechol oxidation [34]. The reaction mixture consisted of 0.5 mL pyrocatechol, 2 mL PBS, and 0.5 mL extract. The absorbance at 410 nm was recorded at initial mixture and after 2 min of mixture, respectively. One unit of PPO was calculated as a change of 0.01 in absorbance per min based on the fresh weight of fruit samples.

The PAL (phenylalanine ammonia lyase) activity was measured based on the lysis of l-phenylalanine [35]. The reaction mixture consisted of 2 mL PBS, 1 mL extract, and 1 mL l-phenylalanine (0.02 M). The mixture was incubated in boiled water for 1 h. After this, 0.2 mL HCl (6 M) was added to terminate the reaction. Then, the absorbance at 290 nm was recorded before and after a water bath. One unit of PAL was calculated as a change of 0.01 in absorbance per hour based on the fresh weight of fruit samples.

### 2.11. Measurement of Metabolites

Approximately 2 g pepper fruit samples around inoculated sites were taken to determine the content of flavonoids and total phenols. Fruit samples were ground with cold 1% HCl (dissolved in methanol), adding distilled water to obtain a total volume of 20 mL. After the reaction at 4 °C in darkness for 20 min, the mixture was centrifuged at 12,000 rpm for 20 min. We collected the supernatant to determine the flavonoid and total phenol content using a plant total phenol test kit (A143-1-1) and plant flavonoids test kit (A142-1-1), respectively (Nanjing Jiancheng Bioengineering Institute, Nanjing, China) [36]. 

### 2.12. Data Analysis

All data were presented as the mean ± SD (standard deviation) of at least three replicates. Significant differences between two treatments (*p* < 0.05 or *p* < 0.01) were compared using ANOVA (one-way analysis of variance) with an *F* test. Significant differences (*p* < 0.05) among multiple treatments were compared using LSD (least significant difference test). The package “corrplot” in R was applied to perform Pearson correlation analysis [37]. TBtools was used to generate heatmaps for hierarchical cluster analysis [38].

## 3. Results

### 3.1. Mycelial Growth

*B. cinerea* was cultured on PDA plates containing CA for 48 h to evaluate the growth of mycelia in vitro. CA at 0.1–0.8 mM inhibited the radical growth of mycelia significantly in an dose-dependent manner (Figure 1A,B). The EC_50_ of CA was 0.5 mM based on the linear regression of the colony diameter shown in Figure 1B. The MIC (minimal inhibitory concentration) of CA against the radical growth of *B. cinerea* was 1.1 mM. A time-course experiment was performed to monitor the growth of mycelia upon CA exposure at 0.5 mM. CA (0.5 mM) began to inhibit mycelial growth significantly after 12 h. The average mycelial growth speed for the control and CA treatment was 0.108 and 0.055 cm/h, respectively (Figure 1C).

As CA at 0.5 mM resulted in a decrease in mycelial growth by 50%, we selected CA at 0.2, 0.5, and 0.8 mM as low, median, and high concentrations, respectively, to investigate the antifungal effect of CA against *B. cinerea* in detail. Compared to the control group, the mycelial fresh weight significantly decreased by 40.0%, 60.9%, and 83.7% upon CA addition at 0.2, 0.5, and 0.8 mM, respectively (Figure 2A). The mycelial dry weight remarkably decreased by 33.5%, 61.0%, and 92.6% upon CA addition at 0.2, 0.5, and 0.8 mM, respectively, compared to the control (Figure 2B). These results suggested that CA exposure remarkably decreased the biomass of *B. cinerea*.

### 3.2. CA Induced Mycelial Injury of B. cinerea

The morphology of mycelia was observed under SEM after treatment for 48 h (Figure 3). The control group without CA showed smooth and intact mycelia. CA at 0.2 mM induced distorted mycelia with minor collapse on the surface. CA at 0.5 mM resulted in much more distorted and collapsed mycelia, with fracture and shrinkage. Treatment with CA at 0.8 mM led to the almost complete breakdown of the mycelia, which shrank severely with the loss of surface integration. 

To confirm the CA-induced collapse of mycelia, several physiological parameters in CA-treated mycelia were investigated. CA treatments enhanced the mycelial relative conductivity (Figure 4A), suggesting the leakage of intracellular matter. The lipid peroxidation of the plasma membrane can be indicated by the TBARS content. CA elevated the TBARS content in a dose-dependent manner (Figure 4B), indicating the occurrence of lipid peroxidation in mycelial cells upon CA exposure. CA also induced glycerol accumulation in mycelia (Figure 4C), suggesting the occurrence of an osmotic response in CA-treated mycelia.

ROS frequently attack lipids in the membrane to lead to lipid oxidation. A specific fluorescent probe, DCFH-DA, was used to detect endogenous ROS in mycelia. CA-treated mycelia showed extensive DCF fluorescence, suggesting the accumulation of ROS (Figure 5). Mycelial cell death was labeled with specific fluorescent probe PI. CA at 0.2 mM induced slight PI fluorescence in mycelia. The PI fluorescence was more extensive with the increase in CA concentration, suggesting that CA treatment led to mycelial death (Figure 5).

### 3.3. CA Damaged the Spores of B. cinerea

CA inhibited the production of spores in a dose-dependent manner. Spore numbers decreased by 35.0%, 65.0%, and 78.3% significantly upon CA addition at 0.2, 0.5, and 0.8 mM, respectively (Figure 6A). CA also inhibited the germination of healthy spores. After culturing for 12 h, almost all of the spores germinated in the control group. However, only 54.0%, 29.7%, and 12.0% of spores germinated upon CA exposure at 0.2, 0.5, and 0.8 mM, respectively (Figure 6B). 

Flow cytometry was applied to detect DCF-labeled spores. The number of DCF-labeled spores increased under CA treatment as compared to the control (Figure 7A), suggesting that CA induced ROS accumulation in spores. The increase in PI-labeled spores also suggested that CA induced the cell death of spores (Figure 7B).

### 3.4. Hierarchical Cluster and Correlation Analysis of Physiological Parameters in CA-Treated B. cinerea

Hierarchical cluster analysis was applied to compare the changes in the parameters upon CA treatment. One cluster consisted of the parameters related to fungal growth and development, such as the mycelial diameter, mycelial fresh weight, mycelial dry weight, spore number, and spore germination. All these parameters decreased compared to the control in a dose-dependent manner (Figure 8A), suggesting the inhibitory effect of CA against *B. cinerea* growth. Glycerol content, TBARS content, and relative conductivity were clustered together. These three parameters increased with the increase in CA concentration as compared to the control (Figure 8A), suggesting that membrane injury occurred in CA-treated *B. cinerea*.

Pearson correlation analysis was performed for all of the measured parameters. The injury indexes (glycerol content, TBARS content, and relative conductivity) were negatively correlated with the growth indexes (such as mycelial diameter, mycelial fresh weight, mycelial dry weight, spore number, and spore germination) (Figure 8B).

### 3.5. CA Suppresses the Infection of B. cinerea on Pepper Fruit

The above results confirmed the capability of CA in inhibiting *B. cinerea* in vitro. We further evaluated the possible role of CA in pepper fruits infected with *B. cinerea*. Inoculation with *B. cinerea* resulted in severe gray mold on pepper fruits, such as enlarged browning and decay around inoculated sites (control in Figure 9A). Pepper fruits pretreated with CA (0.5 mM) showed very slight disease symptoms (the right-hand fruit in Figure 9A). The average size of lesions was significantly decreased by 68.7% on CA-treated fruits as compared to the control (Figure 9B).

The defensive responses of pepper fruits after 12 h of *B. cinerea* inoculation were evaluated by measuring the activity of several defensive enzymes. *B. cinerea* inoculation induced a significant increase in POD activity compared to the control (neither CA nor *B. cinerea* inoculation). Pretreatment with CA induced a remarkable increase in POD activity on fruits as compared to *B. cinerea* inoculation alone (Figure 10B,C). The changes in PPO and PAL activity were similar to those of POD activity under *B. cinerea* and CA treatment. 

Inoculation with *B. cinerea* significantly decreased the content of flavonoids and total phenols in pepper fruits. Pretreatment with CA effectively enhanced the content of these metabolites as compared to *B. cinerea* inoculation alone (Figure 10D,E).

## 4. Discussion

Applying plant extracts is an effective approach to control gray mold on vegetables and fruits [39]. CA is a small-molecule phytochemical with broad antimicrobial activity. CA can be applied as a natural food preservative [40]. In this study, we found that CA effectively controlled the gray mold on fresh pepper fruits by showing antifungal activity against *B. cinerea* and inducing defensive responses in pepper fruits, which in turn maintained pepper quality.

It has been reported that CA shows potential antifungal activity against variable phytopathogenic fungi, such as *Fusarium moniliforme*, *Verticillium fungicola*, *Sclerotinia homoeocarpa*, *Trichophyton rubrum*, and *Aspergillus fumigates* [41,42]. The antifungal activity of CA has been associated with the disruption of cell wall integrity in *Geotrichum citri-aurantii* [43]. CA damaged the mycelia of *B. cinerea* by inducing cell collapse and death. The loss of plasma membrane integrity may be one of the most important reasons, evidenced by ion leakage from mycelial cells. Plasma membrane damage is a typical mechanism behind the fungicidal activity of CA against *Alternaria alternata* [44]. The inhibition of ergosterol biosynthesis is involved in the antifungal activity of CA against *Fusarium sambucinum* [20], but we found that ROS accumulation may be another important reason. CA-induced ROS accumulation can cause lipid peroxidation, resulting in membrane damage by increasing membrane permeability in mycelial cells of *B. cinerea*. The triggering ROS accumulation is an important action of CA-induced bacterial cell death. CA-induced apoptosis-like death can be linked to the triggering of ROS accumulation in *Microcystis aeruginosa* [30]. Nox (NADPH oxidase) is an important enzyme generating ROS in eukaryotes [45]. Nox-produced ROS contributes to the cell leakage in fungi [46]. Besides damaging the cell membrane, ROS can act on mitochondrial proteins to induce fungal death [47]. CA can induce mitochondrial dysfunction in human leukemia K562 cells [48]. Therefore, further studies are needed to investigate whether and how CA triggers the Nox-ROS system to damage cell membranes and mitochondria in *B. cinerea* mycelia. 

Glycerol accumulation is a typical consequence of osmotic stress in pathogenic fungi [49]. CA induced glycerol accumulation in *B. cinerea* mycelia, indicating that osmotic stress occurred. Filamentous fungi adapt to osmotic stress mainly through the HOG (high osmolarity glycerol) pathway, which consists of a set of sensors and transducers activating the biosynthesis of glycerol [50]. Fungicidal activity has been associated with osmotic stress by targeting the HOG pathway [51]. A conserved HOG pathway has been found in *B. cinerea* as well [52]. It is of interest to further understand whether the modulation of the HOG pathway contributes to the antifungal activity of CA against *B. cinerea*.

It was reported that CA-inhibited spore germination resulted in the suppression of mycelial development in *Aspergillus flavus* [40]. The antifungal activity of CA against *B. cinerea* was also related to the inhibition of sporulation and spore viability. *B. cinerea* produces large amounts of spores to finish the life cycle in order to achieve survival, dispersion, and infection among host plants [53]. Here, we found that CA significantly inhibited *B. cinerea* spore generation. This may have been due to the detrimental effect of CA on the vegetative growth of mycelia, weakening the ability of *B. cinerea* to finish its life cycle. CA also inhibited the germination of healthy spores produced by *B. cinerea*. Stress-inhibited sporulation and germination result from ROS accumulation and oxidative injury in *B. cinerea* [54]. ROS accumulation in *B. cinerea* spore may be one of the possible reasons for CA-suppressed spore viability. 

Fresh pepper fruits pretreated with CA before *B. cinerea* inoculation showed decreased lesion development. This result suggested that CA may induce defense responses in pepper fruits against the invasion of *B. cinerea*. This hypothesis could be partially confirmed by the result that CA increased the activity of defensive enzymes (POD, PAL, and PPO) in pepper fruits. All of these enzymes play roles in helping host plants against pathogens [55,56]. CA induced the accumulation of flavonoids and total phenols in pepper fruits as well. These metabolites are not only important non-enzymatic components of plant defense, but also vital nutrients reflecting the quality of pepper fruits [56,57]. At present, little is known about CA-regulated plant physiology. We previously found that CA conferred cadmium tolerance by triggering endogenous Ca^2+^ in plants [58]. In addition, CA can induce defense responses in citrus fruit against the infection of *Geotrichum citri-aurantii* [21]. Moreover, CA can modulate the immune responses in fish against fungal infection [59,60]. These reports and our current results suggest an important role of CA in triggering host immunity to combat biotic or abiotic stress. However, the detailed mechanisms need further study.

Several reports suggest that CA derivatives also have effective (or even enhanced) antimicrobial activity [61,62,63]. In addition, antifungal agents can be encapsulated in nanoparticles in order to enhance the antifungal effect or to control the release of ingredients, which is proposed as a novel approach to control *B. cinerea* infection [64]. Encapsulation can control the release of essential oils to enhance their efficiency, which can help to decrease the dosage and possible impact of volatile flavors after treatment. Several types of encapsulated nano-CA have been developed to test the bioactivity and ingredient release [65,66]. Therefore, further efforts are needed to design and modify the structure of CA, which would help to apply CA in the biocontrol of *B. cinerea*-caused postharvest gray mold on vegetables and fruits during the supply chain in different scenarios.

## 5. Conclusions

In this study, we revealed that CA was capable of controlling postharvest gray mold on fresh pepper fruit by inhibiting *B. cinerea* and triggering defense responses in pepper. CA with a concentration of more than 0.2 mM effectively inhibited the growth of *B. cinerea* in vitro. The antifungal activity of CA against *B. cinerea* was closely related to ROS accumulation and oxidative injury in both mycelia and spores. Pretreatment with CA triggered defensive responses in pepper, which further prevented the infection of *B. cinerea*. The application of CA in real scenarios needs further study, but our current results extend our knowledge of the antifungal activity of CA as well as the potential application of CA in the preservation of vegetables and fruits.

## Figures and Tables

**Figure 1 foods-12-03458-f001:**
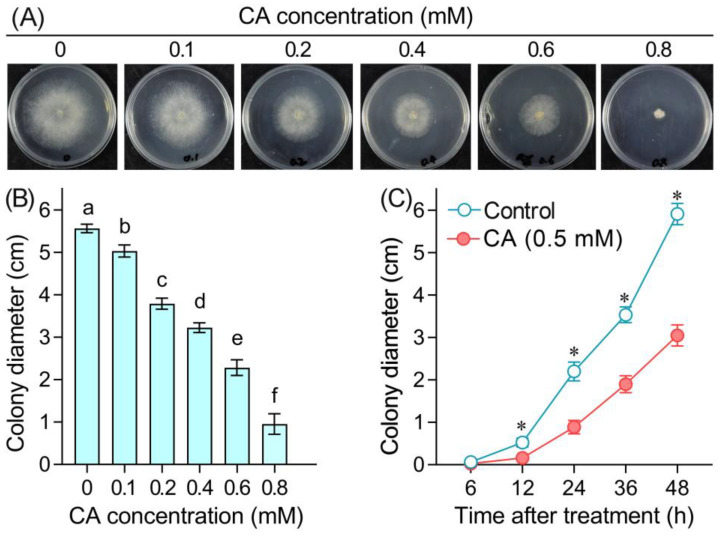
The antifungal effect of CA against the radical growth of *B. cinerea*. (**A**) A photograph of *B. cinerea* growing on PDA plate containing CA at different concentrations. (**B**) The colony diameter of *B. cinerea* on PDA plate. (**C**) The time-course changes in the colony diameter of *B. cinerea* under 5 mM CA treatment. Different lowercase letters in (**B**) indicate significant differences among different treatments (LSD, ANOVA, *n* = 12, *p* < 0.05). The asterisk in (**C**) indicates significant difference between control and CA treatment at each time point (ANOVA, *n* = 8, *p* < 0.05).

**Figure 2 foods-12-03458-f002:**
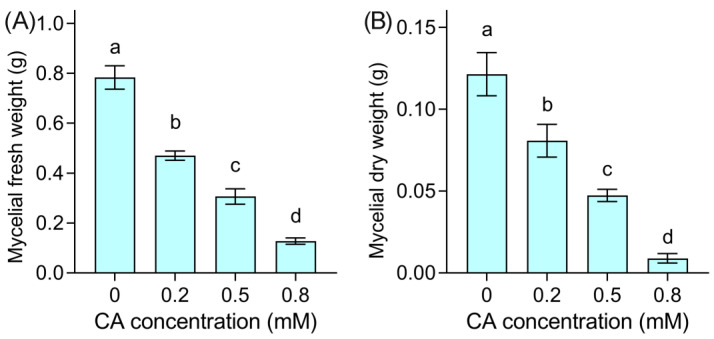
The effect of CA on the mycelial growth of *B. cinerea*. (**A**) Mycelial fresh weight. (**B**) Mycelial dry weight. Different lowercase letters indicate significant differences among different treatments (LSD, ANOVA, *n* = 3, *p* < 0.05).

**Figure 3 foods-12-03458-f003:**
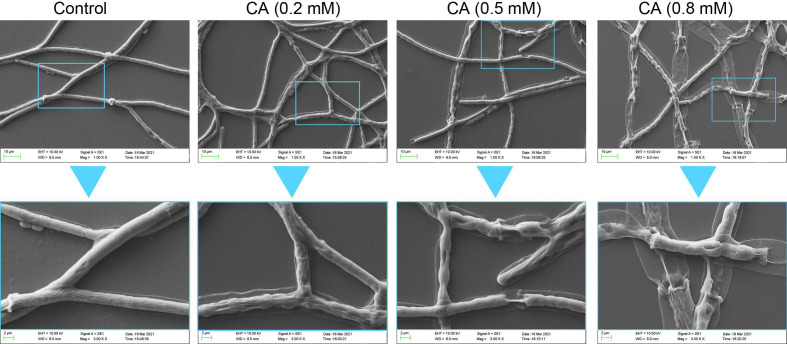
The effect of CA on the mycelial structure of *B. cinerea*. Upper panel and lower panel indicate the magnification of 1000× and 3000×, respectively.

**Figure 4 foods-12-03458-f004:**
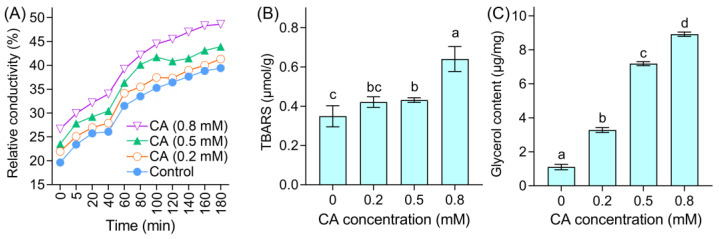
The effect of CA on (**A**) relative conductivity, (**B**) TBARS content, and (**C**) glycerol content in the mycelia of *B. cinerea*. Different lowercase letters in (**B**,**C**) indicate significant differences among different treatments (LSD, ANOVA, *n* = 3, *p* < 0.05).

**Figure 5 foods-12-03458-f005:**
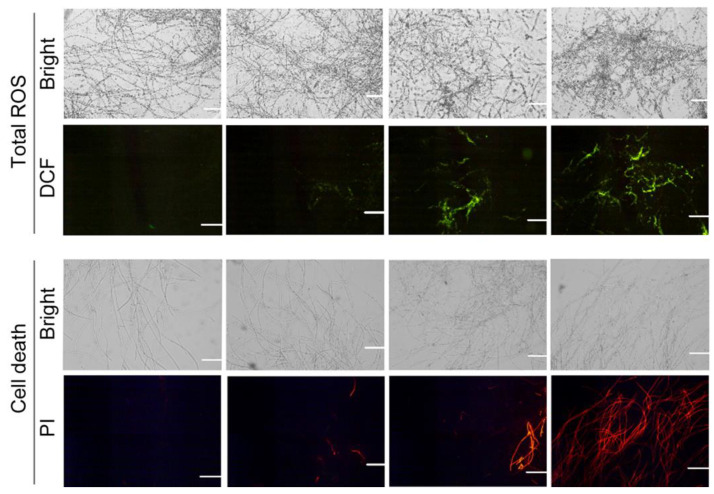
The effect of CA on ROS accumulation and cell death in the mycelia of *B. cinerea*.

**Figure 6 foods-12-03458-f006:**
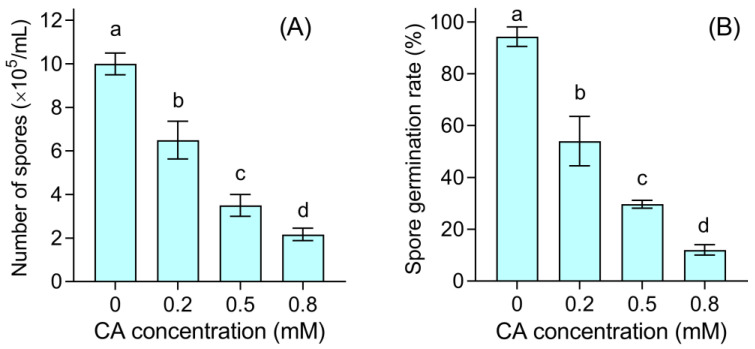
The effect of CA on spore generation and spore germination of *B. cinerea*. (**A**) Number of spores produced by *B. cinerea* upon CA treatment. (**B**) The germination rate of spores upon CA treatment. Different lowercase letters indicate significant differences among different treatments (LSD, ANOVA, *n* = 3, *p* < 0.05).

**Figure 7 foods-12-03458-f007:**
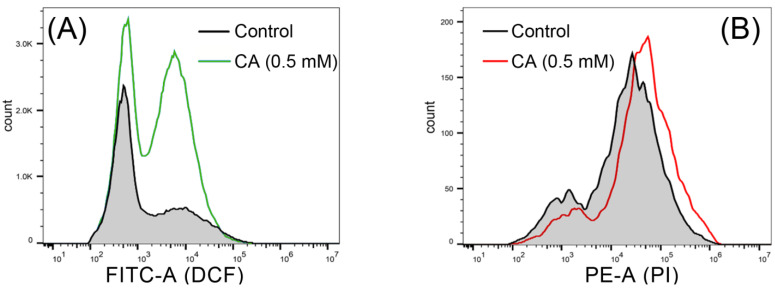
The effect of CA on ROS accumulation and cell death of the spores of *B. cinerea*. (**A**) DCF-labeled spores detected using flow cytometry. (**B**) PI-labeled spores detected using flow cytometry.

**Figure 8 foods-12-03458-f008:**
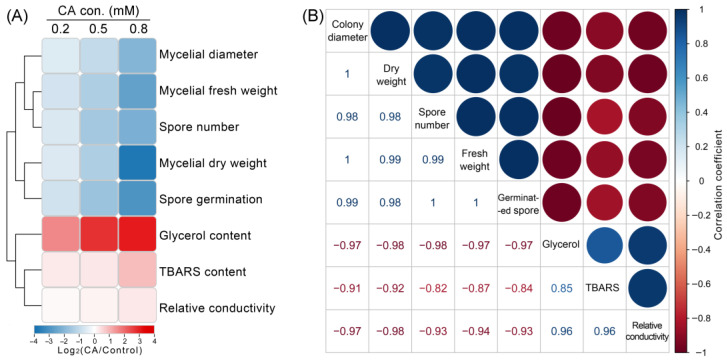
Cluster analysis and correlation analysis among determined physiological parameters. (**A**) Hierarchical cluster analysis of the parameters. For each parameter, the value from CA treatment is presented as log2 (fold change vs. control). (**B**) Pearson correlation analysis among different parameters.

**Figure 9 foods-12-03458-f009:**
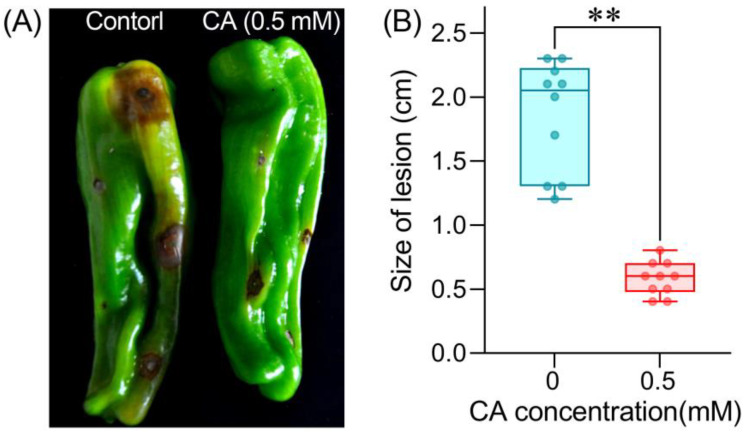
The effect of CA on the infection of *B. cinerea* in pepper fruits. (**A**) A photograph of the lesions on pepper fruits. (**B**) The size of lesions measured on pepper fruits. Double asterisks indicate significant differences between treatments (ANOVA, *n* = 10, *p* < 0.01).

**Figure 10 foods-12-03458-f010:**
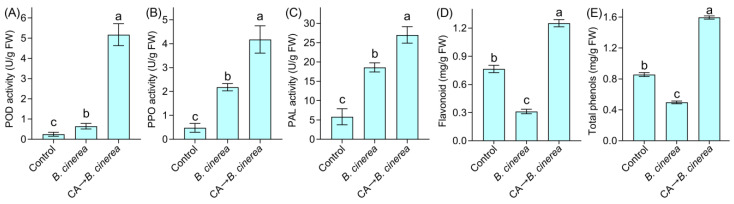
The effect of CA on the defensive enzyme activity and metabolites in pepper fruits infected with *B. cinerea*. (**A**) POD activity. (**B**) PPO activity. (**C**) PAL activity. (**D**) Flavonoid content. (**E**) Total phenol content. Different lowercase letters indicate significant differences among treatments (ANOVA, *n* = 3, *p* < 0.05).

## Data Availability

Not applicable.

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
