# Peer review of "Cinnamaldehyde Inhibits Postharvest Gray Mold on Pepper Fruits via Inhibiting Fungal Growth and Triggering Fruit Defense"

_foods, 2023, doi:10.3390/foods12183458_

Round 1
Reviewer 1 Report
line 18: please write the exact doses and times used in the treatment
line 27: invasion change to infection
28: pepper is a vegetable you don't need to write vegetables and fruits.
40: Pepper is a widely cultivated vegetables - should be Peppers is a widely cultivated vegetables
68-71: too many phrases, please write clear, and as much as you can short sentences. too many not necessary words
74 : first the origin of Botrytis, it was monoculture. PCR identified?
74: PDA manufacturer?
75: PDB same as 74 line
80: CA obtained from where? what form?
80: not clear. You grew Botrytis for 2 days and only then added CA? how added ?
80: what concentrations (0.1-0.8), do you mean 0,1, 0,2, 0,3, 0,4, 0,5, 0,6 and so on???
82: why the different concentration than in the 80 line?
81: how measured? reference?
no references in 74-86 lines, replications?
83: what measurements
how much time the experiment took?
have you evaluated the growth rate?
and the third experiment where 83-86 lines?
not clear what was done
88: how collected
121: variety? Where from obtained? Where they sprayed pesticides? why 25 C storage temperature?
123: add photo or scheme of inoculation, in which parts of fruit were inoculated? replications?
124: dark or light?
125: how was treated CA? sprayed? dipped?
126: it should've inoculated with PDA without fungi
128: have you evaluated the fungi growth diameter cm ?
150: reference?
169: no info in methods that measured after 48 h
170: in vitro italic
172: how is EC50 evaluated? add to methods
177: time course not mentioned in methods
177: figure 1B, 0 - is control? in 1C is the control
170: radical growth of mycelia - add to methods, reference
175: growth speed - how measured, reference
187: mycelial dry weight no info in method, add how measured, reference
183: how you can select 0.2, 0.5, and 0.8 mM if above you evaluated 0,2, 0,5 and 0,8. where from you get 0,5 results? how do you know its medium?
176: colony diameter - add to methods, reference
175: fresh weight significantly - how measured? reference
difference between fresh and dry weight? references in methods?
189: biomass - how evaluated?
196: smooth and intact mycelia - add to methods, evaluation scale, reference
same 198: distorted and collapsed mycelia - add to methods, evaluation scale, reference
213: relative conductivity, (B) TBARS content, and (C) glycerol content 213 in the mycelia of B. cinerea. - add to methods and reference
217: specific fluorescent probe DCFH-DA - no in methods
212: why time evaluation different (0-180 min)? no in methods info about time
176: time after treatment 6-48 h, and in figure 4 0-180 min. why different timing?
203: what time frame was evaluated?
221: mycelial death - reference, how evaluated? method
225: spores in a dose-dependent manner - method, reference
223: time frame, after what time was evaluated ROS?
226: what time were cultured spores? add to methods
not clear methods, no references,
232: spore germination- reference, methods
236: flow cytometry - method, reference
244: Hierarchical cluster analysis - add to methods, reference
265: in vitro. italic
274: how are lesions of infection measured? method?
276: defensive responses how are measured? method
290: add some more discussion of CA on other fungi, crops,
no clear conclusion, what concentration and time to use overall?
maybe with different storage temperatures results would be different? why
Author Response
Responses to Reviewer #1
- line 18: please write the exact doses and times used in the treatment
Response: Thanks for your suggestion. We have provided the details here in revised manuscript.
- line 27: invasion change to infection.
Response: Thanks for your suggestion. We have changed “invasion” to “infection” here in revised manuscript.
- 28: pepper is a vegetable you don't need to write vegetables and fruits.
Response: Thanks for your suggestion. We have changed “vegetables and fruits” to “peppers” here in revised manuscript.
- 40: Pepper is a widely cultivated vegetables - should be Peppersis a widely cultivated vegetables
Response: Thanks for your suggestion. We have changed “pepper” to “peppers” here in revised manuscript.
- 68-71: too many phrases, please write clear, and as much as you can short sentences. too many not necessary words
Response: Thanks for your suggestion. We have rephrased this paragraph to make it clear in revised manuscript.
- 74 : first the origin of Botrytis, it was monoculture. PCR identified?
Response: Thanks for your comment. We have provided the origin of the monoculture of B. cinerea in revised manuscript.
- 74: PDA manufacturer?
75: PDB same as 74 line
80: CA obtained from where? what form?
Response: Thanks for your comment. We have provided all the required information about PDA, PDB, and CA in revised manuscript.
- 80: not clear. You grew Botrytis for 2 days and only then added CA? how added ?
Response: Thanks for your comment. CA was added to PDA before culturing B. cinerea. We have provided this information here in revised manuscript.
- 80: what concentrations (0.1-0.8), do you mean 0,1, 0,2, 0,3, 0,4, 0,5, 0,6 and so on???
Response: Thanks for your comment. We have provided the detailed concentration of CA here in revised manuscript.
- 82: why the different concentration than in the 80 line?
Response: Thanks for your comment. CA at 0.1-0.8 mM were used for the determination of the sensitivity of B. cinerea to CA. This result helped us calculate the EC50 at 0.5 mM. Here we used 0.2, 0.5, and 0.8 as low, median, and high CA concentration, respectively, to studied the physiological and biochemical responses of B. cinerea. We have provided these information in revised manuscript.
- 81: how measured? reference?
Response: Thanks for your comment. The colony diameter was measured by using a ruler crossing the center of the colony circle. We have provided the method and reference here in revised manuscript.
- no references in 74-86 lines, replications?
Response: Thanks for your comment. The references for the methods in this part have been supplied in revised manuscript.
- 83: what measurements
Response: Thanks for your comment. This is for the measurement for the germination rate for the spores. It has been supplied in revised manuscript.
- how much time the experiment took?
Response: Thanks for your comment. The mycelial growth experiment was conducted for 48 h. The spore germination rate was conducted for 12 h.
- have you evaluated the growth rate?
Response: Thanks for your comment. The growth rate of mycelia has been evaluated, which can be found in the first part of “Results” in revised manuscript.
- and the third experiment where 83-86 lines? not clear what was done
Response: Thanks for your comment. This experiment was conducted to evaluated the inhibitory effect of CA on the germination rate of spores. We have supplied this information in revised manuscript.
- 88: how collected
Response: Thanks for your comment. The mycelia were collected by filtration of culture solution with double gauze followed by washing with phosphate buffer solution. The method has been supplied here in revised manuscript.
- 121: variety? Where from obtained? Where they sprayed pesticides? why 25 C storage temperature?
Response: Thanks for your comment. The variety and origin of the pepper have been provide here in revised manuscript. The peppers were grown in a greenhouse without using any pesticides. The 25℃ was selected as the storage temperature because it is suitable for the growth of B. cinerea.
- 123: add photo or scheme of inoculation, in which parts of fruit were inoculated? replications?
Response: Thanks for your comment. A schematic model for mycelial inoculation has been supplied in supplementary material as Figure S2. Five fruits were inoculated for each treatment, which has been provided in revised manuscript.
- 124: dark or light?
Response: Thanks for your comment. The inoculated fruits were kept in a light chamber with photoperiod of 12 h and active radiation of 200 μmol/(m2 s). This information has been supplied here in revised manuscript.
- 125: how was treated CA? sprayed? dipped?
Response: Thanks for your comment. CA was sprayed on fruit surface. This information has been supplied here in revised manuscript.
- 126: it should've inoculated with PDA without fungi
Response: Thanks for your comment. Actually, the control group was inoculated with B. cinerea, but sprayed with water instead of CA.
- 128: have you evaluated the fungi growth diameter cm ?
Response: Thanks for your comment. Yes, we measured the fungi growth as the size of disease lesion in Figure 9.
- 150: reference?
Response: Thanks for your comment. The reference has been supplied here in revised manuscript.
- 169: no info in methods that measured after 48 h
Response: Thanks for your comment. The method and the reference has been supplied in revised “Materials and Methods”.
- 170: in vitro italic
Response: Thanks for your comment. It has been revised to intalic.
- 172: how is EC50 evaluated? add to methods
Response: Thanks for your comment. The method and the reference have been supplied in revised “Materials and Methods”.
- 177: time course not mentioned in methods
Response: Thanks for your comment. It has been supplied in revised “Materials and Methods”.
- 177: figure 1B, 0 - is control? in 1C is the control
Response: Thanks for your comment. In Figure 1B, CA at 0 mM is the control.
- 170: radical growth of mycelia - add to methods, reference
Response: Thanks for your comment. The method and reference have been supplied in revised “Materials and Methods”.
- 175: growth speed - how measured, reference
Response: Thanks for your comment. It has been supplied in revised “Materials and Methods”.
- 187: mycelial dry weight no info in method, add how measured, reference
Response: Thanks for your comment. It has been supplied in revised “Materials and Methods”.
- 183: how you can select 0.2, 0.5, and 0.8 mM if above you evaluated 0,2, 0,5 and 0,8. where from you get 0,5 results? how do you know its medium?
Response: Thanks for your comment. The 0.5 mM is EC50 calculated from the linear regression of colony diameter under CA at 0.1, 0.2, 0.4, 0.6, and 0.8 mM. Then we selected 0.2, 0.5, and 0.8 mM as low, median, and high concentration, respectively. These information have been supplied in revised “Materials and Methods”.
- 176: colony diameter - add to methods, reference
Response: Thanks for your comment. It has been supplied in revised “Materials and Methods”.
- 175: fresh weight significantly - how measured? Reference
Response: Thanks for your comment. It has been supplied in revised “Materials and Methods”.
- difference between fresh and dry weight? references in methods?
Response: Thanks for your comment. The fresh weight is the weight of mycelia measured after harvesting immediately. Then the mycelia were dried in an oven to evaporate all the water inside of mycelial cells in order to get the completely dried biomass. It has been supplied in revised “Materials and Methods”.
- 189: biomass - how evaluated?
Response: Thanks for your comment. The biomass was evaluated based on the fresh weight and dry weight of mycelia. It has been supplied in revised “Materials and Methods”.
- 196: smooth and intact mycelia - add to methods, evaluation scale, reference
Response: Thanks for your comment. The structure of mycelia was evaluated based on morphological analysis according to previous publication. The method and reference have been supplied in revised “Materials and Methods”.
- same 198: distorted and collapsed mycelia - add to methods, evaluation scale, reference
Response: Thanks for your comment. It has been supplied in revised “Materials and Methods”.
- 213: relative conductivity, (B) TBARS content, and (C) glycerol content 213 in the mycelia of B. cinerea. - add to methods and reference
Response: Thanks for your comment. The methods and references can be found in 2.4, 2.5, and 2.6 in the section of “Materials and Methods”.
- 217: specific fluorescent probe DCFH-DA - no in methods
Response: Thanks for your comment. The methods and references can be found in 2.8 in the section of “Materials and Methods”.
- 212: why time evaluation different (0-180 min)? no in methods info about time. 176: time after treatment 6-48 h, and in figure 4 0-180 min. why different timing?
Response: Thanks for your comment. Therefore, the 0-180 min is only for the method during the determination of conductivity, but not the treatment time (48 h). This has been explained in revised “Materials and Methods”.
- 203: what time frame was evaluated?
Response: Thanks for your comment. The morphology was determined after treatment for 48 h. We have supplied this information here in revised manuscript.
- 221: mycelial death - reference, how evaluated? Method
Response: Thanks for your comment. Dead cells but not live cells can be labeled by PI. More PI fluorescence indicate more dead cells. The method and reference can be found in 2.8 in revised “Materials and Methods”.
- 225: spores in a dose-dependent manner - method, reference
Response: Thanks for your comment. It has been supplied in 2.3 in revised “Materials and Methods”.
- 223: time frame, after what time was evaluated ROS?
Response: Thanks for your comment. ROS was evaluated after treatment with CA for 48 h. It has been in 2.8 in revised “Materials and Methods”.
- 226: what time were cultured spores? add to methods
Response: Thanks for your comment. The methods including the time have been added in 2.3 in revised “Materials and Methods”.
- 232: spore germination- reference, methods
Response: Thanks for your comment. It has been added in 2.3 in revised “Materials and Methods”.
- 236: flow cytometry - method, reference
Response: Thanks for your comment. It has been added in 2.8 in revised “Materials and Methods”.
- 244: Hierarchical cluster analysis - add to methods, reference
Response: Thanks for your comment. The reference for this method can be found in 2.12 in revised “Materials and Methods”.
- 265: in vitro. Italic
Response: Thanks for your comment. It has been changed to italic here.
- 274: how are lesions of infection measured? method?
Response: Thanks for your comment. The lesion is almost a circle. For each lesion, the diameter was evaluated by measuring 3 times from different directions crossing the center of the lesion. It has been added in 2.9 in revised “Materials and Methods”.
- 276: defensive responses how are measured? Method
Response: Thanks for your comment. The defensive responses were evaluated by measuring the activity of several defensive enzymes. We have supplied this information here in revised manuscript. The method for the determination of enzymatic activity can be found in 2.10 in revised “Materials and Methods”.
- 290: add some more discussion of CA on other fungi, crops,
Response: Thanks for your comment. We have provided more discussion about it in revised manuscript.
- no clear conclusion, what concentration and time to use overall?
Response: Thanks for your comment. We have rephrased the part of conclusion in revised manuscript.
- maybe with different storage temperatures results would be different? Why
Response: Thanks for your comment. We think that it would dependent on the fungal growth at different storage temperatures. Low temperature would slow down the spore germination and mycelial growth. In this scenario, CA at low concentrations would be enough to control B. cinerea. Things would be different at high temperature that can promote spore germination and mycelial growth. Thus CA at high dose may be needed to control B. cinerea effectively. In addition, high temperature can accelerate the respiration of fruits, which may change the interaction between host defense and pathogenic fungi. And the role of CA in the induction of fruit defense would be re-evaluated. This is another complex topic to be studied further.
Reviewer 2 Report
Major revisions are needed
foods-2571191-peer-review-v1
Introduction
The authors need to state the temperature and RH ranges at which the B. cinerea grow at.
It will be beneficial for readers to state the chemical structure of Cinnamaldehyde. And why certain elements in their structure are beneficial as natural postharvest treatments for fungal suppression.
What are the management tools already used to manage the B.C in fresh produce during storage?
How about fungicide, Controlled Atmosphere, Modified Atmosphere, heat treatments? Neither of these techniques was listed in the introduction. You must justify your work as a novel piece of science by listing the deficiency in previous work.
Materials and Methods
You must provide references for measurements in sections 2.1 to 2.9.
Results
The sizes of figures are not proportional to the axis’ titles. The sizes of the figures need to be larger.
No conclusions?
Author Response
Responses to Reviewer #2
Introduction
- The authors need to state the temperature and RH ranges at which the B. cinerea grow at.
Response: Thanks for your comment. We have provided this information in revised “Introduction”.
- It will be beneficial for readers to state the chemical structure of Cinnamaldehyde. And why certain elements in their structure are beneficial as natural postharvest treatments for fungal suppression.
Response: Thanks for your suggestion. We have supplied the chemical structure of cinnamaldehyde in supplementary Figure S1. And the relationship between the chemical structure and antifungal activity has been supplied in revised “Introduction”.
- What are the management tools already used to manage the B.C in fresh produce during storage? How about fungicide, Controlled Atmosphere, Modified Atmosphere, heat treatments? Neither of these techniques was listed in the introduction. You must justify your work as a novel piece of science by listing the deficiency in previous work.
Response: Thanks for your comment. We have provided the information of the approaches for managing postharvest gray mold in revised “Introduction”.
Materials and Methods
- You must provide references for measurements in sections 2.1 to 2.9.
Response: Thanks for your comment. We have supplied all the missing references in this part.
- Results: The sizes of figures are not proportional to the axis’ titles. The sizes of the figures need to be larger.
Response: Thanks for your comment. We have revised some figures in order to get a proportional size in revised manuscript.
- No conclusions?
Response: Thanks for your comment. We have supplied a “5. Conclusion” part after “4. Discussion” in revised manuscript.
Reviewer 3 Report
The manuscript entitled “Cinnamaldehyde Inhibits Postharvest Gray Mold on Pepper Fruits via Inhibiting Fungal Growth and Triggering Fruit Defense” was reviewed.
The manuscript describes an interesting topic and results show clearly the usefulness of using cinnamaldehyde to reduce the grow of Botrytis cinerea in pepper fruits.
However, a more precise determination of the minimal inhibitory concentration should be attended.
None
Author Response
The manuscript entitled “Cinnamaldehyde Inhibits Postharvest Gray Mold on Pepper Fruits via Inhibiting Fungal Growth and Triggering Fruit Defense” was reviewed.
The manuscript describes an interesting topic and results show clearly the usefulness of using cinnamaldehyde to reduce the grow of Botrytis cinerea in pepper fruits.
However, a more precise determination of the minimal inhibitory concentration should be attended.
Response: Thanks for your comment. Actually, we have determined the MIC (minimal inhibitory concentration) of CA against B. cinerea. The MIC is 1.1 mM. We have provided this result in the part of “Results” in revised manuscript.
Reviewer 4 Report
The manuscript entitled “Cinnamaldehyde Inhibits Postharvest Gray Mold on Pepper Fruits via Inhibiting Fungal Growth and Triggering Fruit Defense” presents the cinnamaldehyde (CA) effects of B. cinerea on fresh pepper fruit in vitro. The study is interesting and the manuscript is well structured and presented. Also, the study is supported by the results presented, but some changes should be made.
- The role of the Abstract is to highlight what is most important. The Abstract should be modified and put into evidence the importance of using CA, e.g. environmentally friendly natural compounds, medicinal activities, antifungal and antibacterial activity, etc.
- What is the role of CuSO4 at 0.05 g/l concentration?
- Why at the beginning of the study the chosen concentrations were 0, 0.1, 0.2, 0.4, 0.6, and 0.8 mM and then you chose 0, 0.2, 0.5, and 0.8 mM ?
- The discussions are well presented, but a short conclusion section must be introduced in the manuscript.
- What was the intraday and interday precision for each type of measurement?
- English language and typing corrections must be made.
English language and typing corrections must be made
Author Response
Responses to Reviewer #4
- The role of the Abstract is to highlight what is most important. The Abstract should be modified and put into evidence the importance of using CA, e.g. environmentally friendly natural compounds, medicinal activities, antifungal and antibacterial activity, etc.
Response: Thanks for your suggestion. We have revised the abstract in order to emphasize the importance of this study and CA.
- What is the role of CuSO4 at 0.05 g/l concentration?
Response: Thanks for your comment. The role of CuSO4 in determining the glycerol content is to react with glycerol to produce dark blue cop-per-glycerol complex with specific absorbance at 630 nm. We have provided this information in revised “Materials and Methods”.
- Why at the beginning of the study the chosen concentrations were 0, 0.1, 0.2, 0.4, 0.6, and 0.8 mM and then you chose 0, 0.2, 0.5, and 0.8 mM ?
Response: Thanks for your comment. In the beginning, we chose CA at 0, 0.1, 0.2, 0.4, 0.6, and 0.8 mM in order to test the sensitivity of B. cinerea. This result helped us calculate the EC50 (median effective concentration) of CA at 0.5 mM. Next, we selected CA at 0.2, 0.5, and 0.8 mM that could cause light, median, and severe stress response in B. cinerea, respectively, which helped evaluate the physiological responses B. cinerea to CA exposure. We have explained this in revised “Materials and Methods” and “Results”.
- The discussions are well presented, but a short conclusion section must be introduced in the manuscript.
Response: Thanks for your suggestion. We have supplied a “5. Conclusion” part after “4. Discussion” in revised manuscript.
- What was the intraday and interday precision for each type of measurement?
Response: Thanks for your comment. For the measurement of each physiological parameter, all the samples were collected after treatment followed by the measurement immediately. And we finished the measurement for each parameter within one day. The data was significantly evaluated by using F-test or LSD with ANOVA comparison. It’s hard to calculate the intraday and interday precision for biological experiments. However, all the experiments have been repeated three to four times, showing similar changes among different treatments.
- English language and typing corrections must be made.
Response: Thanks for your suggestion. We have double checked the whole manuscript to correct all the writing mistakes.
Round 2
Reviewer 1 Report
thank you for corrections